# Identification of Drug Targets and Their Inhibitors in *Yersinia pestis* Strain 91001 through Subtractive Genomics, Machine Learning, and MD Simulation Approaches

**DOI:** 10.3390/ph16081124

**Published:** 2023-08-09

**Authors:** Hamid Ali, Abdus Samad, Amar Ajmal, Amjad Ali, Ijaz Ali, Muhammad Danial, Masroor Kamal, Midrar Ullah, Riaz Ullah, Muhammad Kalim

**Affiliations:** 1Department of Biosciences, COMSATS University Islamabad, Park Road, Tarlai Kalan, Islamabad 44000, Pakistan; 2Department of Biochemistry, Abdul Wali Khan University, Mardan 23200, Pakistan; sabdus591@gmail.com (A.S.); amar.ajmal2022@gmail.com (A.A.); danialshah353@gmail.com (M.D.); masroorkamal007@gmail.com (M.K.); 3Faculty of Biological Sciences, Department of Biochemistry, Quaid-i-Azam University, Islamabad 45320, Pakistan; amjadtanoli10@gmail.com; 4Centre for Applied Mathematics and Bioinformatics (CAMB), Gulf University for Science and Technology, Hawally 32093, Kuwait; ali.i@gust.edu.kw; 5Department of Biotechnology, Shaheed Benazir Bhutto University Sheringal, Dir Upper 18050, Pakistan; drmidrarullah@gmail.com; 6Department of Pharmacognosy, College of Pharmacy King Saud University, Riyadh 11451, Saudi Arabia; rullah@ksu.edu.sa; 7Department of Microbiology and Immunology, Wake Forest School of Medicine, Winston-Salem, NC 27101, USA; mkalim@houstonmethodist.org; 8Houston Methodist Cancer Center/Weill Cornel Medicine, Houston, TX 77030, USA

**Keywords:** subtractive genomics, *Yersinia pestis*, machine learning algorithms, docking, MD simulation

## Abstract

*Yersinia pestis*, the causative agent of plague, is a Gram-negative bacterium. If the plague is not properly treated it can cause rapid death of the host. Bubonic, pneumonic, and septicemic are the three types of plague described. Bubonic plague can progress to septicemic plague, if not diagnosed and treated on time. The mortality rate of pneumonic and septicemic plague is quite high. The symptom-defining disease is the bubo, which is a painful lymph node swelling. Almost 50% of bubonic plague leads to sepsis and death if not treated immediately with antibiotics. The host immune response is slow as compared to other bacterial infections. Clinical isolates of *Yersinia pestis* revealed resistance to many antibiotics such as tetracycline, spectinomycin, kanamycin, streptomycin, minocycline, chloramphenicol, and sulfonamides. Drug discovery is a time-consuming process. It always takes ten to fifteen years to bring a single drug to the market. In this regard, in silico subtractive proteomics is an accurate, rapid, and cost-effective approach for the discovery of drug targets. An ideal drug target must be essential to the pathogen’s survival and must be absent in the host. Machine learning approaches are more accurate as compared to traditional virtual screening. In this study, k-nearest neighbor (kNN) and support vector machine (SVM) were used to predict the active hits against the beta-ketoacyl-ACP synthase III drug target predicted by the subtractive genomics approach. Among the 1012 compounds of the South African Natural Products database, 11 hits were predicted as active. Further, the active hits were docked against the active site of beta-ketoacyl-ACP synthase III. Out of the total 11 active hits, the 3 lowest docking score hits that showed strong interaction with the drug target were shortlisted along with the standard drug and were simulated for 100 ns. The MD simulation revealed that all the shortlisted compounds display stable behavior and the compounds formed stable complexes with the drug target. These compounds may have the potential to inhibit the beta-ketoacyl-ACP synthase III drug target and can help to combat *Yersinia pestis*-related infections. The dataset and the source codes are freely available on GitHub.

## 1. Introduction

*Yersinia pestis* (*Y. pestis*), the causative agent of plague, is a Gram-negative bacterium that was initially extracted by Alexandre Yersin in Hong Kong. The plague impacted human life through a number of pandemics that were originally spread from Central Asia to Africa and Europe [1]. During the last 150 years, plague reached every continent. Plague is present in America, Asia, and Africa in the 21st century [2]. If plaque is not properly treated it can cause rapid death of the host, because *Y. pestis* exhibits a very unusual collection of virulence factors that enable effective flea infection and immune response subversion in mammalian hosts. Bubonic, pneumonic, and septicemic are the three types of plague [3]. Apart from respiratory droplets that cause pneumonic plague and flea bites that cause bubonic plague, gastrointestinal plague is caused by the consumption of uncooked contaminated meat. Bubonic plague can progress to septicemic plague, if not diagnosed and treated on time. The mortality rate of pneumonic and septicemic plague is quite high [4]. Regarding bio-terrorism concerns plague is interesting, and in fact has a long history of use as a biowarfare agent [2]. The symptom-defining disease is the bubo which is a painful lymph node swelling. Almost 50% of bubonic plague leads to sepsis and death if not treated. In rats, humans, and non-human primates, the bubonic plague disease is very common, and the host immune response is slow as compared to other bacterial infections [5]. *Y. pestis* exhibits an extremely distinctive combination of virulence factors that enable effective flea infection and disruption of immunological responses in mammalian hosts, resulting in rapid host death if not treated adequately. Pathogen-associated molecular patterns (PAMPs), iron capture systems, the Yersinia outer membrane proteins (Yops), the broad-range protease Pla, and other virulence factors play crucial roles in the molecular strategies used by *Y. pestis* to evade the human immune system and allow unrestricted bacterial replication in lymph nodes (bubonic plague) and the lungs (pneumonic plague). For diagnostic purposes, some of these immunogenic proteins including the capsular antigen F1 are used [2].

Clinical isolates of *Y. pestis* showed resistance to many antibiotics such as tetracycline, spectinomycin, kanamycin, streptomycin, minocycline, chloramphenicol, and sulfonamides. Microbial pathogen genome sequencing projects have grown rapidly in the last ten years [6]. It always takes ten to fifteen years to bring a single drug to the market [7].

A bacterial pathogen is responsible for many illnesses. A large number of people die from these diseases every year, and several resistant strains have evolved which makes these diseases difficult to control. For the design of new antibiotics against drug-resistant pathogens, new drug targets are required. However, drugs currently being used for the treatment of infections may have fewer or more side effects in humans. Repetitive use has also contributed to the evolution of drug resistance in pathogens. In this regard, subtractive genomics is a novel strategy for the identification of new drug targets in disease-causing bacteria [8]. An ideal drug target must be essential to the pathogen’s survival and should be absent in the host. In silico subtractive proteomics is an accurate, rapid, and cost-effective approach for the discovery of drug targets [9], and also provides genome information if the host and pathogen proteomes are available [10]. In this study, we predicted drug targets via a subtractive genomics approach and then applied machine learning approaches to predict the active hits against the drug target. Furthermore, the active hits were docked against the drug target. A computer simulation technique known as molecular dynamics (MD) explores the physical movements of atoms and molecules. In molecular dynamics (MD), atoms and molecules interact for a fixed period of time resulting in the system’s evolution over time. A total of three best docking score compounds along with the standard drug were simulated for 100 ns. 

## 2. Results and Discussion

### 2.1. Subtractive Genomics

Figure 1 displays the strategy for the identification of the drug target against the *Y. pestis* 91001 strain and the number of proteins shortlisted in each step. The entire proteome of the *Y. pestis* [11] 91001 strain (ASM788v1) with 1870 sequences was downloaded from the NCBI database www.ncbi.nlm.nih.gov (accessed on 28 January 2023) in the FASTA format. For paralogous protein recognition, CD-hit was used, which is a commonly used database for the clustering of genes/proteins to minimize redundancy during the drug target identification process [12]. After applying the CD-hit tool, out of 1870 sequences 197 were found as paralogous and were removed. In order to avoid cross-reactivity of drugs with a host protein, the non-paralogous sequences were submitted to BLASTp against the *Homo sapiens* proteome and sequences that showed similarities to the human host were excluded. In this case, a total of 314 sequences were excluded as they showed similarity with the human host. 

Genes that are necessary for an organism’s survival, growth, and adaptability are considered essential genes. These genes have a common function and are mostly preserved and evolving across taxa. Thus, to evaluate the essential genes of *Y. pestis*, we utilized the Essential Gene Database (DEG). The Essential Gene Database (DEG) is an online database that contains experimentally validated information about important genes of bacteria, fungi, plants, and animals [13]. This database is constantly modified and contains all the genes that are considered essential for basic cell survival. For antibacterial drugs, essential proteins are known as excellent targets [14]. To evaluate the essential genes of *Y. pestis,* the remaining human non-homologous sequences with a threshold of 10–5 were subjected to BLASTp against the DEG, which revealed 380 proteins as essential to the pathogen. These sequences involve functional, non-functional, or uncharacterized proteins, and have been deemed viable to the life cycle of the pathogen.

KEGG is an invaluable resource that may help to explain essential biological processes, their resources, and functions in the cell, organism, and overall ecosystem. The KEGG database includes the full description of a network of metabolic pathways. It helps in the identification of sequences that are important in metabolism. KEGG determines the potential target for a drug based on the specific metabolism of the pathogen. Analysis of metabolic pathways for the essential protein sequences was carried out by the use of the KEGG database, which revealed 24 specific pathways in the *Y. pestis* 91001 strain. The specific pathways as well as their IDs are shown in Table 1. A total of sixteen proteins and their particular pathways are shown in Table 2.

It is important to identify the subcellular position of proteins for their categorization as drug or vaccine targets. Membrane proteins can be considered as vaccine targets while cytoplasmic can be considered drug targets [15]. A computational tool called CELLO v2.5 enables the prediction of the subcellular localization of the essential and non-homologous protein sequences [16]. Our findings indicate about 80% percent of proteins were in the cytoplasm, 15% were in the inner membrane, and 5% were periplasmic. Druggability analysis revealed a total of 12 proteins as drug targets, as these 12 proteins revealed similarity with the FDA-approved drugs of the drug bank database (Table 3).

Analysis of non-homology with gut flora is essential for the identification of orthologs in gut flora. Around 1014 microorganisms reside in the gastrointestinal tract of healthy humans [17]. Furthermore, a total of four proteins were identified as non-gut flora proteins (Table 4).

As illustrated by recent research [14], subtractive genome methodologies presented here for the *Yersinia pestis* study are currently successfully employed for the identification of drug targets in pathogenic species. In the present study, we predicted four human non-homologous, cytoplasmic, and non-gut flora drug targets: the LysR family transcriptional regulator, exodeoxyribonuclease III, tRNA guanosine-N1-methyltransferase TrmD, and beta-ketoacyl-ACP synthase III in the *Y. Pestis* strain 91001. As part of the type II fatty acid biosynthesis (FASII) system, the beta-ketoacyl-acyl carrier protein synthases, FabB, FabF, and FabH, catalyze the elongation of fatty acids to synthesize components of lipoproteins, phospholipids, and lipopolysaccharides, which are essential for bacterial growth as well as survival. These enzymes are interesting targets for the design of new therapeutics [18]. 

### 2.2. Machine Learning-Based Virtual Screening

By the process of subtractive genomics, four drug targets were identified in the *Y. pestis* 91001 strain. Among the four identified drug targets, the data was available for the beta-ketoacyl-ACP synthase III drug target. Therefore, ML-based virtual screening was performed for the beta-ketoacyl-ACP synthase III drug target. Machine learning algorithms are quick and effective, frequently utilized in domains such as drug discovery, structural biology, and chemo-informatics. These approaches are suitable for the virtual screening of huge compound libraries to categorize molecules as active or inactive or to rank them based on their activity levels, since they can deal with high-dimensional data [19]. Recently, a number of studies have been conducted for the virtual screening of different databases using machine learning approaches [20,21]. We also employed the same approaches for the virtual screening study. A total of 386 compounds were retrieved from the binding databank database “https://www.bindingdb.org/ (accessed on 15 April 2023)” [22]. A total of 500 decoys were generated. By combining the active compounds and decoys, a dataset of 886 compounds was prepared. MOE software was used to calculate the 2D features of all the compounds [20]. A recursive feature selection is a technique that selects a subset of relevant features (columns) from a dataset. A machine learning algorithm with fewer features will run more efficiently with less space or time complexity [23]. A total of 22 optimum features were selected from the dataset, as shown in Figure 2. The scikit-learn python library was used to split the data into a train set (70%) and a test set (30%), as shown in Figure 3.

The two supervised machine learning algorithms KNN and SVM implemented in the scikit-learn library were applied to the dataset to train and test the dataset. The accuracy of KNN was 97%, and that of SVM was 83%. The best k value identified was 7. Other performance evaluation parameters of both the KNN and SVM are described in Table 5. 

We used grid search parameters and 10-fold cross-validation to find the best optimum model, and the procedure was repeated 10 times to obtain reliable results [19]. The ROC-AUC curve for both the KNN and SVM was generated in python, as shown in Figure 4 and Figure 5, respectively. 

A high AUC value indicates the high performance of the model [20]. As the accuracy of the KNN algorithm was high, we used the KNN model for the virtual screening of the South African natural compounds database. Out of the 1012 compounds, a total of 11 compounds were predicted to be active against the beta-ketoacyl-ACP synthase III target. 

### 2.3. Docking Analysis

Two molecules, such as a protein and a small molecule, can interact with one another in a variety of ways during molecular docking. The docking study predicts the intermolecular framework that is established between a small molecule and a protein or between two proteins [24]. The crystal structure of beta-ketoacyl-ACP synthase III PDB ID 4YLT was retrieved from the PDB database. The compounds predicted as active against the drug target were docked in order to predict the interactions between the compounds and the drug target. The compound 4,5-dichloro-1,2-dithiole-3-one was identified as a standard reference inhibitor for beta-ketoacyl-ACP synthase [25] and it was taken as a control compound in the docking study. For every ligand, 10 different conformations were created. Next, all the docked hits with the top-ranked conformations were collected and saved in a database for interaction analysis. The hits were well accommodated inside the beta-ketoacyl-ACP synthase III drug target and formed different interactions with the target active site residues. The docking scores of the hits were good as compared to the standard drug. The docking score of SANC00450 was −7.40. The compound SANC00450 binding mode revealed that it formed a total of five interactions with active site residues such as Arg 36, Trp 32, Arg 151, Gly 208, and Asn 209. The carbon and sulfur atoms of the ligand formed a hydrogen bond donor interaction with the carbonyl oxygen of glycine and arginine. The sulfur and oxygen atoms of the ligand formed two H-acceptor interactions with Arg and Asn, while the 6-ring moiety of the compound formed one pi–H interaction with the Trp 32 of the receptor. The docking score of compound SANC00717 was −7.01 and the carbon atom and 6-ring moiety of the compound formed three H-donor and one pi–H interaction with Ala 245, Gly 208, Arg 36, and Arg 151, respectively. The docking score of compound SANC00247 was −6.54. The carbon atom of the compound formed an H-donor interaction with Gly 208, Asn 209, and Asn 246, and the oxygen atom of the compound formed two H-acceptor interactions with Arg 151 while the 6-ring moiety of the compound formed one pi–H interaction with Arg 151. The docking score of the reference compound was −5.20 and the compound formed two hydrophobic contacts with the receptor. The sulfur atom of the reference compound formed one H-acceptor interaction with Val 141 while the 5-ring of the reference compound formed one pi–H interaction with Ala 145 active site residues. The docking score and 2D structures of all the active compounds are present in Table 6, while Table 7 represents the properties of the compounds. The 3D interactions of the best-scored compounds in comparison with the reference compounds within the active site are shown in Figure 6. 

### 2.4. Post Simulation Analysis

The three best docking-scored compounds as well as the reference compound were further simulated for 100 ns. It is crucial to investigate dynamic stability using simulated trajectories to determine how ligands interact with the receptor [26]. 

#### 2.4.1. RMSD Analysis

A lower RMSD value indicates higher stability of the system, while a higher RMSD value indicates low stability of the system [27]. At 30 ns of simulation, the beta-ketoacyl-ACP synthase/SANC00450 system achieved equilibrium. The RMSD graph demonstrated that the amplitude of the fluctuations increased from 30–65 ns when the RMSD score was 2.5 Å, and then decreased to 2 Å with no fluctuations from 65–100 ns. The RMSD plot of beta-ketoacyl-ACP synthase/SANC00717 revealed small fluctuations during 40–65 ns with an RMSD value of 3 Å; soon after 65 ns, the system converged and achieved stability, until 100 ns with an RMSD value of 2.2 Å. Initially, the RMSD value of beta-ketoacyl-ACP synthase/SANC00417 was 2.5 Å, but after 20 ns very high fluctuations were observed from 20–60 ns with an RMSD value of 3 Å; however, after 60 ns the system achieved stability and no major or minor changes were observed till 100 ns with an RMSD value of 2.5 Å. The RMSD of the control compound in complex with the receptor indicates fluctuations between 20–30 and 50–70 ns, but after 70 ns the system achieved stability with an average RMSD of 3 Å. The RMSD plots of all the systems are present in Figure 7. 

#### 2.4.2. RMSF Analysis

To identify the mobile and fluctuated regions in all four systems, we used per residue root mean square fluctuation (RMSF) analysis. In all the systems, residues 70–79 and 230–240 had higher fluctuations in the calculations. The RMSF plot of all the systems is shown in Figure 8.

#### 2.4.3. Principal Component Analysis and Free Energy Landscape

In order to investigate the dominant dynamic behavior of all the systems, principal component analysis (PCA) was performed. The majority of the combined dominating motions along the trajectory were captured by the first 10 eigenvectors. These 10 eigenvectors were also taken out, and their impact on the overall variation of beta-ketoacyl-ACP synthase III was investigated. Using the first two components (PC1 and PC2), the final PCA figure was generated. The motions that were most likely to be attributable were compared using the 2D plots. The color schemes represent the periodic transitions between conformations (from blue to dark red). These results show that the beta-ketoacyl-ACP synthase/SANC00450 system gained less motion in phase space and clustered into a narrow phase space, indicating the system’s stability (Figure 9). The beta-ketoacyl-ACP synthase/SANC00717system, however, achieved conformational changes but then eventually oscillated back to its initial stable state. This demonstrates the system’s stable behavior. The systems beta-ketoacyl-ACP synthase/SANC00417 and beta-ketoacyl-ACP synthase/control were less compact, and more periodic jumps were observed in these systems, which indicates less stability. Figure 8 shows the PCA plots of all the simulated systems. Moreover, the first two principal components were used to calculate the Gibbs free energy landscape (FEL). The FEL estimated for all systems is shown in Figure 10. The blue color in Figure 10 represents the lowest energy state, while the red color represents the highest energy state. The enriched energy minima for beta-ketoacyl-ACP synthase III/SANC00450, beta-ketoacyl-ACP synthase III/SANC00717, and beta-ketoacyl-ACP synthase III/SANC00247 can been seen to cover a large blue area. Additionally, FEL analysis demonstrated that all the complexes gained minimum energy, revealing the most stable conformations.

#### 2.4.4. Dynamics Cross Correlation Matrix (DCCM)

The inter-residues correlation was analyzed by the dynamic cross-correlation map (DCCM). Strongly correlated (positive) and anti-correlated (negative) motions between the residues are shown in red-yellow and blue-green, respectively, throughout the MD simulation. Figure 11 demonstrates the positive and negative correlation among the residues. Positively correlated movements were particularly noticeable at the active site region, and a higher proportion of pairwise-associated residues represents the stable binding of the compounds with the target protein. 

## 3. Materials and Methods

The overall workflow is presented in Figure 12. 

### 3.1. Pathogen Proteome Retrieval

The entire proteomes of the *Y. pestis* 91001 strain were downloaded from the NCBI database “www.ncbi.nlm.nih.gov (accessed on 28 January 2023)” in the FASTA format.

### 3.2. Paralogs Proteins Identification

For paralogous protein recognition, we used CD-hit, which is a commonly used database for the clustering of genes/proteins to minimize redundancy during the drug target identification process [12]. All sequences were subjected to a CD-hit tool and kept the sequence identification cut-off by 60% to preserve strict standards for eliminating duplicate proteins [28]. The paralog proteins were essentially eliminated from further analysis, and the non-paralogous proteins that had greater than 100 amino acid sequences were taken up. A protein having less than 100 amino acids is generally assumed to be non-essential, and therefore omitted [29]. 

### 3.3. Human Non-Homologous Proteins Identification

To find human non-homologous proteins, BLASTp against the human proteome was performed with a bit score greater than 100 and an expected threshold of 10^−4^. Human homologs were omitted, and for further study, the remaining non-homologous proteins were processed. This step was conducted in order to avoid the cross-reactivity of drugs with a host protein [30]. Proteins for which no hit has been found are referred to as non-homologs.

### 3.4. Finding Essential Genes

The DEG database is an online database that contains experimentally validated information about important genes of bacteria, fungi, plants, and animals [13]. To find the essential genes, BLASTp against DEG was performed using an e-value of 10^10^ and a bit score > 100. The remaining non-essential genes were removed.

### 3.5. Pathways Analysis

For specificity, functional annotation of non-homologous essential proteins was accomplished by KAAS (KEGG Automatic Annotation Server). Using the KEGG database, the metabolic pathways of the host, as well as the pathogen, were analyzed comparatively. For drug target identification, we selected proteins that were involved in the pathogen-specific pathway [31].

### 3.6. Proteins Localization Prediction

Periplasm proteins can be found in microbes at different subcellular locations such as extracellular, cytoplasm, and membranes (both outer membrane and plasma membrane). It is important to identify the subcellular position for their categorization as drug or vaccine targets. Membrane proteins can be considered vaccine targets while cytoplasmic can be considered drug targets [15]. Using the CELLO tool, subcellular locations of proteins were predicted [16].

### 3.7. Virulent Proteins Identification

Bacteria use virulence factors and weaken the mechanism of host defense with the aid of adhesion, colonization, and invasion, thereby causing diseases. The bacteria proteome against the VFDB database was subjected to BLASTp using an E-value of 0.0001 and a cut-off bit score > 100 [32].

### 3.8. Druggability Analysis of Shortlisted Proteins

In the drug bank database, the druggability of the proteins was searched. Using BLASTp against the drug bank database, the shortlisted proteins were screened for an analysis of their druggability potential. The drug bank database includes a variety of protein targets with IDs that are FDA approved. Targets with an E-value < 0.005 and a bit score of >100 were considered as drug targets [33].

### 3.9. Analysis of Non-Gut Flora Proteins

Analysis of non-homology with gut flora is essential for the identification of orthologs in gut flora. Around 1014 microorganisms reside in the gastrointestinal tract of healthy humans [17]. Gut microbiota help to perform the metabolism of humans, can prevent the growth of harmful species within the gut, and aid in food digestion [34]. Blocking the proteins of gut flora can have an adverse effect on the host [35]. Therefore, to avoid these effects, shortlisted proteins were searched for similarity with the intestinal flora proteins using an E-value of 0.0001. 

### 3.10. Machine Learning-Based Approaches for Virtual Screening 

#### 3.10.1. Dataset Preparation

The BindingDB database was used to retrieve a dataset of 386 compounds with inhibitory activity against beta-ketoacyl-ACP synthase. The DUDE “https://dude.docking.org (accessed on 17 April 2023)” database was utilized for the decoy generations [36], which were considered inactive compounds in this study. The dataset was preprocessed, and a class label was added to the dataset. The descriptors were calculated in the MOE V 2016 software. Negative or non-inhibitors were labeled as 0, while positive or active were labeled as 1. The feature selection algorithm played an important role in the ML models. The benefits of a feature selection approach include avoiding noisy, redundant, and less relevant features as well as reducing the computation time. In light of this, we employed a wrapper-based technique known as SVM-RFE. This approach first builds a model by measuring the weights of all features using SVM with default hyper-parameters [37]. 

#### 3.10.2. K-Nearest Neighbor (KNN)

KNN is a simple machine learning classifier that classifies the data on the basis of the majority vote count among the k closest data points, also known as nearest neighbors [38]. To find the nearest neighbors, this classifier uses different distances such as cosine distance, Euclidean distance, and hamming distance [37]. In this study, we used the KNN classifier to train the dataset retrieved from the binding DB database. 

#### 3.10.3. Support Vector Machine (SVM) 

SVM is a supervised machine learning model that is excellent for the recognition of refining patterns in large datasets [37]. SVM can be applied to binary and multiclass classification problems. The kernel trick, which includes the linear, polynomial, sigmoid, and radial base function (RBF), is used by SVM. These kernels categorize the instances in accordance with a target class label and transfer low dimensional space to higher dimensional space [22]. The optimum C value was found as 10. For KNN and SVM model generations we used the scikit-learn python library (v3.6). 

#### 3.10.4. Model Validation

To access the models’ performance, 10-fold cross-validation and a number of various statistical measures including sensitivity, specificity, accuracy, f1 score, and Matthews’ Correlation Coefficient (MCC) were calculated [39].

#### 3.10.5. Virtual Screening and Molecular Docking

The machine learning model with the highest accuracy was used for the virtual screening study. SANCDB “https://sancdb.rubi.ru.ac.za (accessed on 17 April 2023)”, a freely available database of natural products [40], was used for the virtual screening. Among a variety of accessible resources for docking MOE software was chosen because of its user-friendly graphical interface [41]. The 3D structure of the drug target PDB ID 4YLT was uploaded to MOE software, energy minimization was carried out, and protonation was performed. London dG and GBVI/WSA dG scoring functions were used [42]. 

### 3.11. MD Simulation 

To understand macromolecular structure-to-function relationships at the atomic level, MD simulation is a widely used technique. The most frequent application of MD simulation is to assess the stability of protein–ligand complexes [43]. For the MD simulations study, the Amber 20 package was used [44]. The tleap module of Amber was used for the initial system setup. The protein–ligand complexes were soaked a into TIP3P hydrated cubic box, 8 Å in size. For the neutralization of the residual charges of the system, the counter ions were added. The MD systems underwent energy minimization, heating, density equilibration, and equilibration under periodic boundary conditions. The last production step of 100 ns as an NPT ensemble was completed at 310 K. After the 100 ns MD simulation was successfully completed. Root means square deviation (RMSD), root mean square fluctuation (RMSF), and principle component analysis (PCA) were performed to measure the strength of the protein–ligand interaction [45].

## 4. Conclusions

In this study, the subtractive genomics approach was used for the prediction of drug targets in *Yersinia pestis*. Machine learning approaches lead to the identification of eleven compounds active against the beta-ketoacyl-ACP synthase III drug target. Out of eleven compounds, three compounds revealed better stability and interactions as compared to the standard compound. Our predicted hits may have the potential to inhibit beta-ketoacyl-ACP synthase III to combat *Y.pestis*-associated infections. It is further recommended to perform an in vitro and in vivo study on the identified new compounds against *Y. pestis.*

## Figures and Tables

**Figure 1 pharmaceuticals-16-01124-f001:**
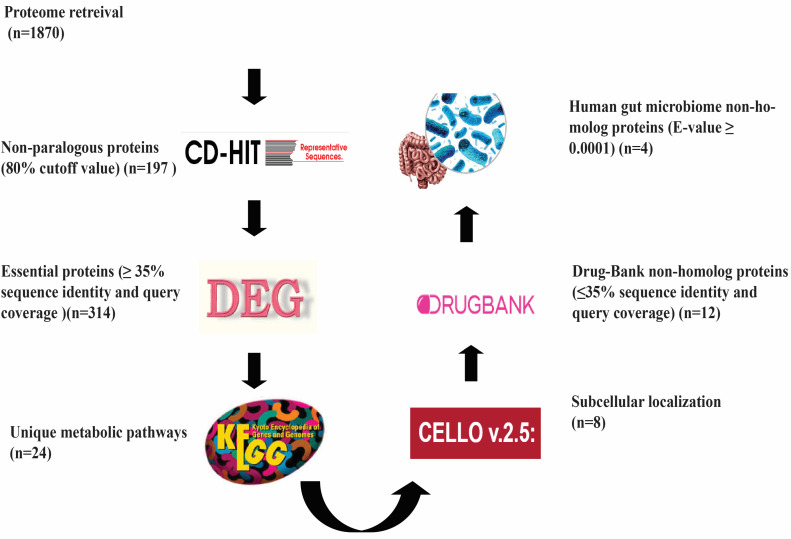
Steps involved in the process of subtractive genomics and the number of proteins shortlisted from each step.

**Figure 2 pharmaceuticals-16-01124-f002:**
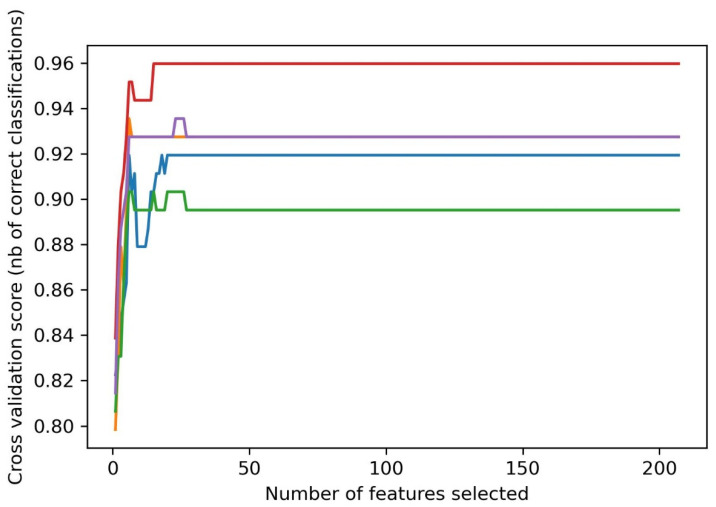
Optimum features selection curve and cross-validation.

**Figure 3 pharmaceuticals-16-01124-f003:**
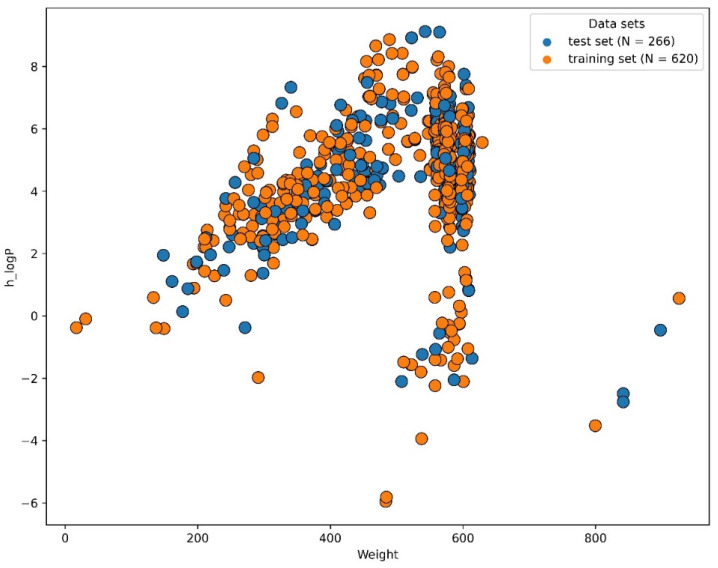
The train and test split of the dataset (70% train and 30% test sets).

**Figure 4 pharmaceuticals-16-01124-f004:**
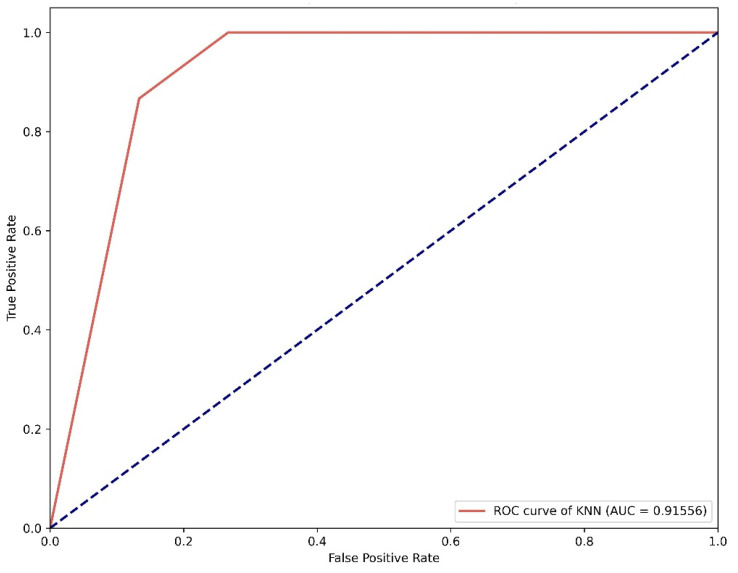
ROC-AUC curve for KNN model indicates an AUC value of 0.91.

**Figure 5 pharmaceuticals-16-01124-f005:**
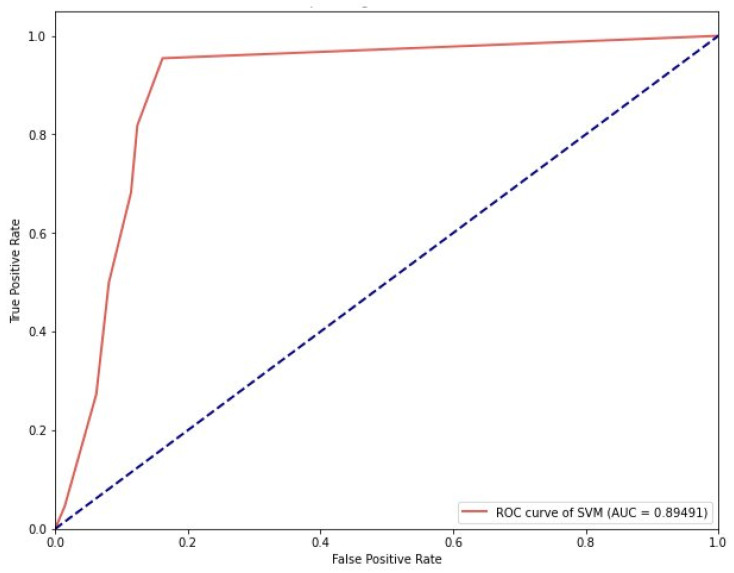
ROC-AUC curve of SVM model indicates an AUC value of 0.89.

**Figure 6 pharmaceuticals-16-01124-f006:**
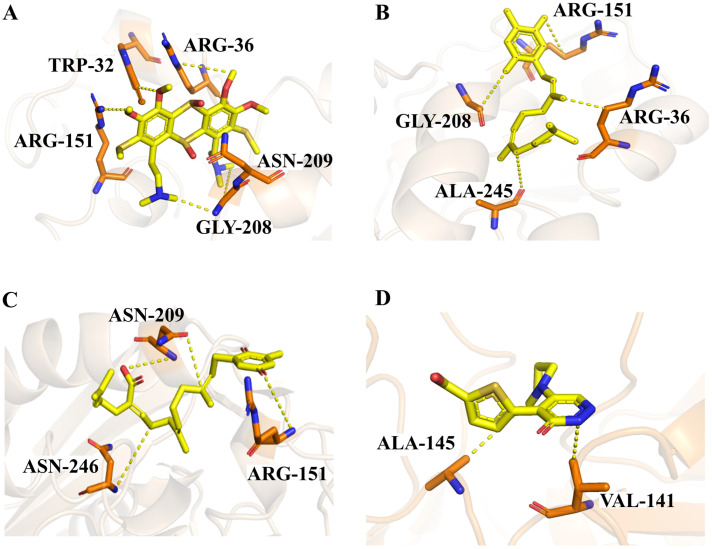
3D interactions of (**A**) SANC00450, (**B**) SANC00717, (**C**) SANC00247, and (**D**) control in complex with beta-ketoacyl-ACP synthase III drug target.

**Figure 7 pharmaceuticals-16-01124-f007:**
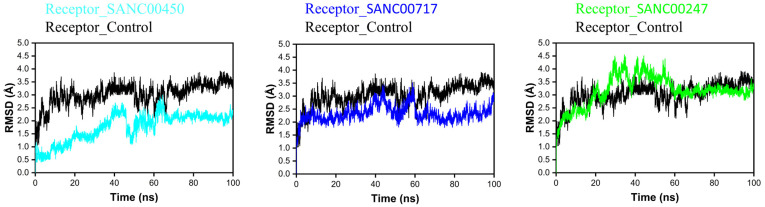
RMSD plot of SANC00450 (cyan), SANC00717 (blue), SANC00247 (green), and control (black) in complex with the beta-ketoacyl-ACP synthase III drug target.

**Figure 8 pharmaceuticals-16-01124-f008:**
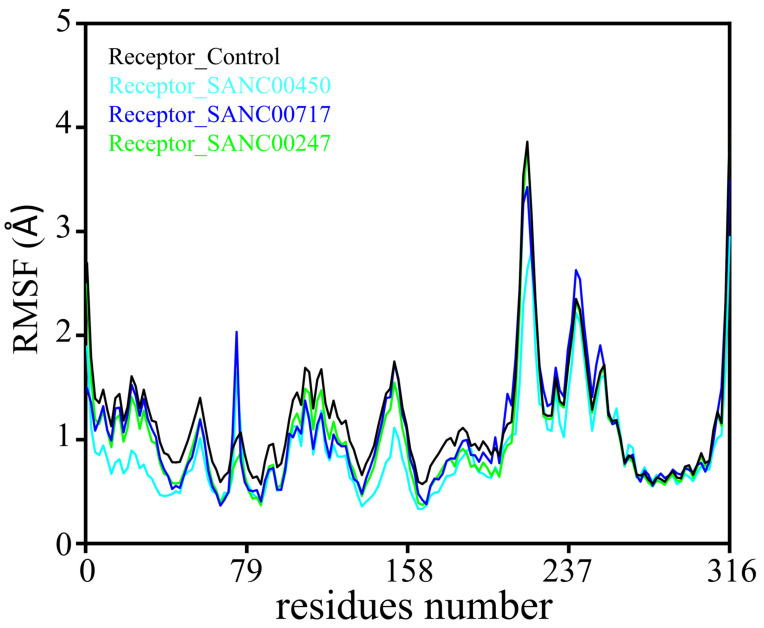
Residual flexibility analysis of SANC00450 (cyan), SANC00717 (blue), SANC00247 (green), and control (black) in complex with the beta-ketoacyl-ACP synthase III drug target.

**Figure 9 pharmaceuticals-16-01124-f009:**
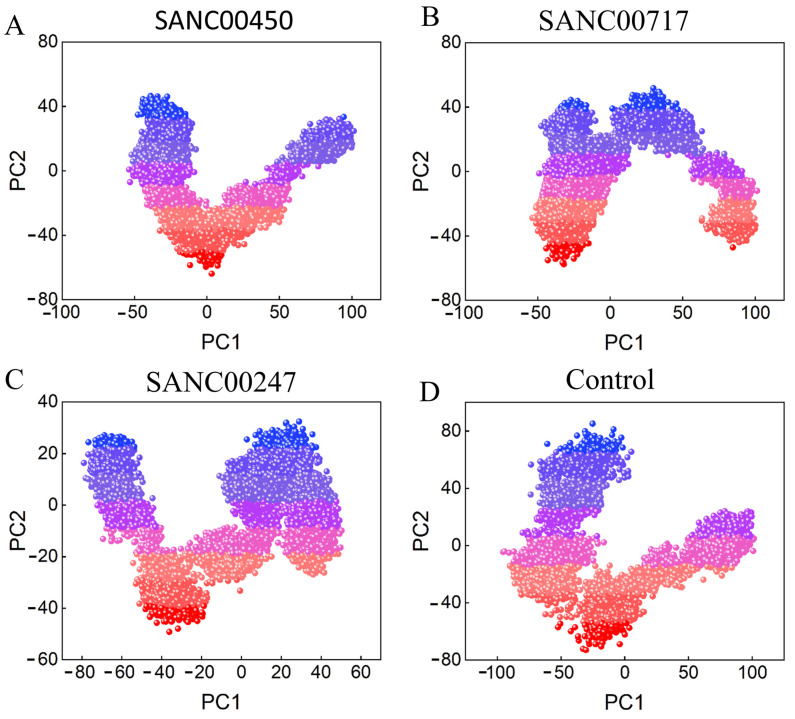
PCA plots of (**A**) SANC00450, (**B**) SANC00717, (**C**) SANC00247, and (**D**) control in complex with beta-ketoacyl-ACP synthase III drug target.

**Figure 10 pharmaceuticals-16-01124-f010:**
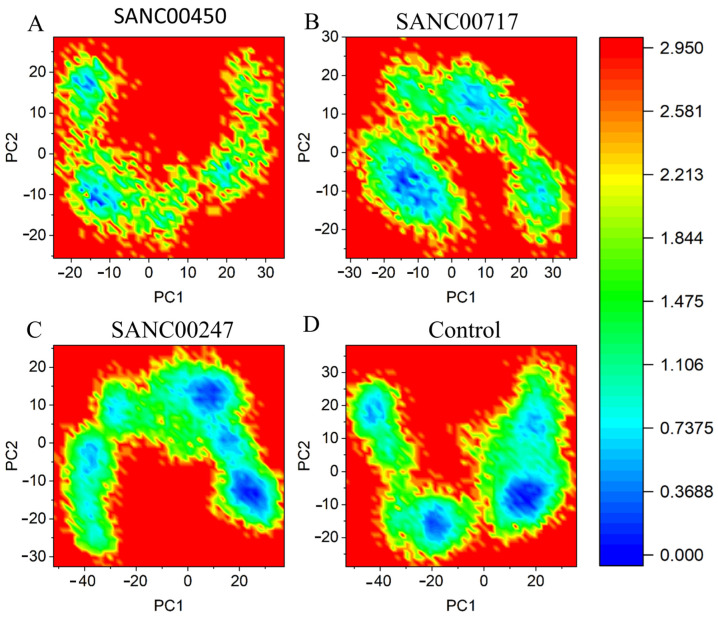
FEL analysis of (**A**) SANC00450 (**B**) SANC00717 (**C**) SANC00247 and (**D**) control in complex with beta-ketoacyl-ACP synthase III drug target. The first two eigenvectors were used for free energy landscape analysis. In each plot the lowest energy conformer is indicated by deep blue color.

**Figure 11 pharmaceuticals-16-01124-f011:**
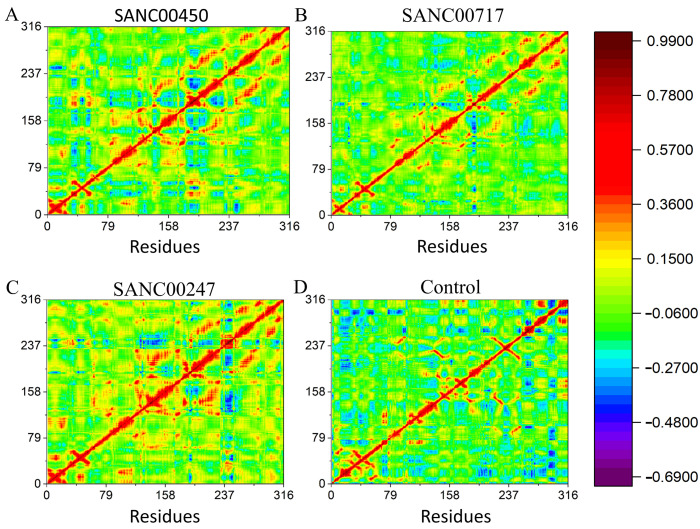
DCCM plots of (**A**) SANC00450, (**B**) SANC00717, (**C**) SANC00247, and (**D**) control in complex with beta-ketoacyl-ACP synthase III drug target. The X and Y axis shows the total number of residues in the receptor.

**Figure 12 pharmaceuticals-16-01124-f012:**
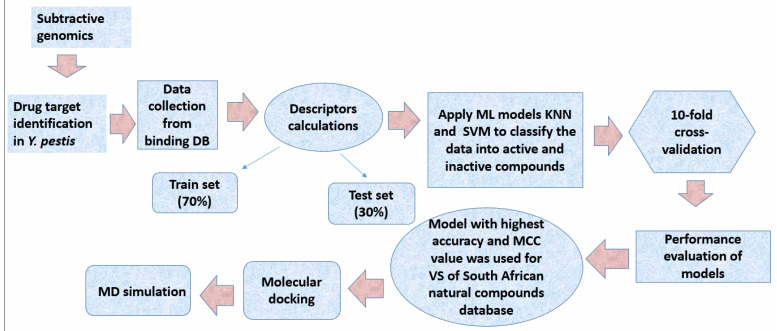
Overall step-by-step workflow of the project.

**Table 1 pharmaceuticals-16-01124-t001:** Specific pathways of pathogen.

S. No	Pathway ID	Pathway Name
1	ypm00281	Geraniol degradation
2	ypm00332	Carbapenem biosynthesis
3	ypm00361	Chlorocyclohexane and chlorobenzene degradation
4	ypm00362	Benzoate degradation
5	ypm00364	Fluorobenzoate degradation
6	ypm00401	Novobiocin biosynthesis
7	ypm00540	Lipopolysaccharide biosynthesis
8	ypm00550	Peptidoglycan biosynthesis
9	ypm00623	Toluene degradation
10	ypm00625	Chloroalkane and chloroalkene degradation
11	ypm00626	Naphthalene degradation
12	ypm00633	Nitrotoluene degradation
13	ypm00930	Caprolactam degradation
14	ypm01054	Nonribosomal peptide structures
15	ypm01120	Microbial metabolism in diverse environments
16	ypm01501	Vancomycin resistance
17	ypm01503	Cationic antimicrobial peptide (CAMP) resistance
18	ypm05135	*Yersinia* infection
19	ypm03070	Bacterial secretion system
20	ypm02020	Two-component system
21	ypm02040	Flagellar assembly
22	ypm02024	Quorum sensing
23	ypm02030	Bacterial chemotaxis
24	ypm02060	Phosphotransferase system (PTS)

**Table 2 pharmaceuticals-16-01124-t002:** Proteins involved in unique pathways.

S. No	Accession No	KO Code	Pathway Name
1	WP_002208960.1	K02988	Bacterial chemotaxis
2	WP_002211667.1	K03070	Bacterial secretion system
3	WP_002224818.1	K07658	Quorum sensing
4	WP_002230619.1	K00349	Yersinia infection
5	WP_002211580.1	K03789	Vancomycin resistance
6	WP_002209578.1	K01736	Peptidoglycan biosynthesis
7	WP_002213325.1	K06958	Quorum sensing
8	WP_002213337.1	K13497	Flagellar assembly
9	WP_002222284.1	K09014	Two-component system
10	WP_002218949.1	K11216	Bacterial chemotaxis
11	WP_011906295.1	K03592	Two-component system
12	WP_002213082.1	K03634	Quorum sensing
13	WP_002211073.1	K01826	Phosphotransferase system (PTS)
14	WP_002210412.1	K19775	Phosphotransferase system (PTS)
15	WP_002209326.1	K03807	Vancomycin resistance
16	WP_002211370.1	K09998	Vancomycin resistance

**Table 3 pharmaceuticals-16-01124-t003:** Druggability potential of the targets.

S. No	Accession No	Drugbank Target	Drugbank ID	Localization
1	WP_002208960.1	drugbank_target|P1588040S ribosomal protein S2	DB09130	Cytoplasmic
2	WP_002211667.1	drugbank_target|P27695 DNA-(apurinic or apyrimidinic site) lyase	DB04967	Cytoplasmic
3	WP_002224818.1	drugbank_target|P13632 C4-dicarboxylate transport transcriptional regulatory protein DctD	DB09462	InnerMembrane
4	WP_002230619.1	drugbank_target|Q9WXS0 Transcriptional regulator, IclR family	DB01942	Periplasmic
5	WP_002211580.1	drugbank_target|O14786 Neuropilin-1	DB00039 DB04895	InnerMembrane
6	WP_002209578.1	drugbank_target|Q52369 Cytochrome c4	DB03754 DB09462	InnerMembrane
7	WP_002213325.1	drugbank_target|P52758 Ribonuclease UK114	DB03793	Cytoplasmic
8	WP_002213337.1	drugbank_target|P0ACQ4 Hydrogen peroxide-inducible genes activator	DB03793	Cytoplasmic
9	WP_002222284.1	drugbank_target|P06971 Ferrichrome-iron receptor	DB03017 DB04160	Cytoplasmic
10	WP_002218949.1	drugbank_target|P12996 Biotin synthase	DB03754	Cytoplasmic
11	WP_011906295.1	drugbank_target|P13632 C4-dicarboxylate transport transcriptional regulatory protein DctD	DB09462	Cytoplasmic
12	WP_002213082.1	drugbank_target|P9WKE1 Thymidylate kinase	DB04160	Cytoplasmic

**Table 4 pharmaceuticals-16-01124-t004:** Analysis with gut flora proteins.

S. No	Accession No	Similarity with Gut Flora
1	WP_002208960.1	No
2	WP_002211667.1	No
3	WP_002213325.1	Yes
4	WP_002213337.1	Yes
5	WP_002222284.1	No
6	WP_002218949.1	Yes
7	WP_011906295.1	No
8	WP_002213082.1	Yes

**Table 5 pharmaceuticals-16-01124-t005:** Performance parameters of the KNN and SVM models.

Model	Accuracy	ROC-AUC Score	F1	MCC
KNN	97%	0.91	0.98	0.97
SVM	83%	0.89	0.78	0.65

**Table 6 pharmaceuticals-16-01124-t006:** The docking score and 2D structures of all the active compounds.

Compound ID	Structure	Docking Score
SANC00450	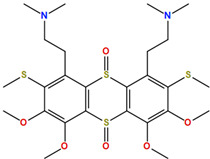	−7.40
SANC00717	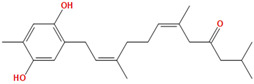	−7.01
SANC00247	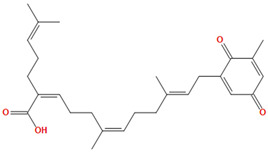	−6.54
SANC00269	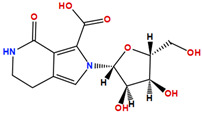	−6.50
SANC00451	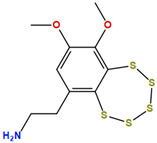	−6.20
SANC00728	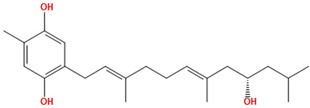	−6.14
SANC00258	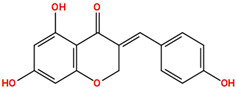	−5.64
SANC00735	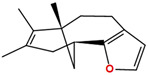	−4.38
SANC01025	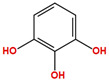	−4.08
SANC01023	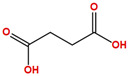	−4.04
SANC00129	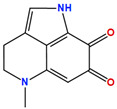	−5.24
4,5-dichloro-1,2-dithiole-3	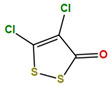	−5.20

**Table 7 pharmaceuticals-16-01124-t007:** Drug-likeness of the active compounds.

Compound ID	Weight	H-Donor	H-Acceptor	Logp	Toxicity
SANC00450	602.86	0	6	3.90	No
SANC00717	340.49	2	3	5.84	No
SANC00247	424.58	1	4	7.86	No
SANC00269	312.28	5	7	1.10	No
SANC00451	339.55	1	3	2.03	No
SANC00728	346.51	3	3	5.0	No
SANC00258	284.27	3	5	2.47	No
SANC00735	216.32	0	0	4.12	No
SANC01025	125.11	3	3	0.35	No
SANC01023	118.09	2	4	-0.77	No
SANC00129	202.21	1	2	1.32	No

## Data Availability

Data is contained within the article.

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
