# Peer review of "Identification of Drug Targets and Their Inhibitors in Yersinia pestis Strain 91001 through Subtractive Genomics, Machine Learning, and MD Simulation Approaches"

_pharmaceuticals, 2023, doi:10.3390/ph16081124_

Round 1

Reviewer 1 Report (Previous Reviewer 2)

Authors have considered my points and improved the manuscript substantially. Although some things still lacking, but sufficient for publication in Pharmaceuticals as this journal is not computer science related and going very deep in ML is not in it's scope. 

Fine

Author Response

Authors have considered my points and improved the manuscript substantially. Although some things still lacking, but sufficient for publication in Pharmaceuticals as this journal is not computer science related and going very deep in ML is not in it's scope. 

Substantial changes has been made in the manuscript and all the points raised by Reviewer were answered accordingly.

Reviewer 2 Report (Previous Reviewer 3)

Dear Authors,

Your present article has been reviewed,

This work deserves attention since it highlights a very important and new topic related to the use of different in silico skills and technics merged with Artificial Intelligence (AI) for the prediction of drug targets and their inhibitors in Yersinia pestis (Y. persitis).

The article is well presented and written in English language. Kindly find below some of my remarks (Majors and Minors).

01- The Introduction section should start at Line 81 instead of line 85.

02- In the Introduction, first word, Authors are invited to put the name of the bacteria "Yersinia pestis" followed by (Y. pestis).

03- Authors are invited to add a clear definition of the Term "MD" in the Introduction.

Best Regards,

Dear Authors,

Some sentences needs to be clearer

Author Response

Reviewer 2

Your present article has been reviewed,

This work deserves attention since it highlights a very important and new topic related to the use of different in silico skills and technics merged with Artificial Intelligence (AI) for the prediction of drug targets and their inhibitors in Yersinia pestis (Y. persitis).

The article is well presented and written in English language. Kindly find below some of my remarks (Majors and Minors).

  • The Introduction section should start at Line 81 instead of line 85.

Reply: According to reviewer suggestions, the introduction is now started at line 81 instead of line 85 as highlighted in the revised manuscript.

  • In the Introduction, first word, Authors are invited to put the name of the bacteria "Yersinia pestis" followed by ( pestis).

Reply: According to the kind reviewer suggestions, in the introduction part the first-word "Yersinia pestis" is now followed by (Y. pestis) as highlighted in the revised manuscript.

  • Authors are invited to add a clear definition of the Term "MD" in the Introduction.

Reply: According to the reviewer suggestions a clear definition of the Term "MD" is now added and highlighted in the introduction. 

This manuscript is a resubmission of an earlier submission. The following is a list of the peer review reports and author responses from that submission.

Round 1

Reviewer 1 Report

Manuscript pharmaceuticals-2409532  entitled “Identification of drug targets and their inhibitors in Yersinia pestis strain 91001 through Subtractive genomics, machine learning, and MD simulation approaches” is lacking in quality, data and research details. The work should be supplemented and re-submitted.

Abstract, “. In spite of the use of high-performance

technology and synthetic chemistry, it always takes ten to fifteen years to bring a drug to the market.” What is high-performance technology? Too broad… rephrase

Abstract, “. As the accuracy of machine learning approaches is very fast and more

accurate as compared to traditional virtual screening, therefore,” Accuracy is fast? Please lecture the complete text with a native English speaker to improve the language in the main text.

Abstract, “The MD simulation revealed all the short-listed compounds have strong stability and binding affinity for the beta keto acyl ACP synthase III drug target” If mentioned in the abstract – specific affinity calculation approaches must be mentioned.

Section 2.1. Please provide the links so the reader can check what was the input data. Then please provide references to the Uniprot and Yersinia pestis Uniprot genome ID. Then please describe in the whole context what part and how the proteome was constructed. Also reference primary literature on Y. pestis proteome: Parkhill J., Wren B.W., Thomson N.R., Titball R.W., Holden M.T.G., Prentice M.B., Sebaihia M., James K.D., Churcher C.M., Barrell B.G. Nature 413:523-527 (2001)

Section 2.1 Essential Gene Database (DEG) does not contain Y. pestis. You just checked the similarity to proteins present in DEG generally and retrieved 380 proteins in Y. pestis that are similar to essential proteins in DEG from all organisms? How was KEGG analysis done on 380 proteins? Provide a flowchart diagram that describes each step in protein identification pipeline with each step showing input and output no. of proteins.

Section 2.1 “It is important to identify the subcellular position of proteins for their categorization

as drug or vaccine targets. Membrane proteins can be considered as vaccine targets while

cytoplasmic can be considered drug targets [12]”  Not true especially for drug development. Why is vaccine development included in this novel antibacterial research work ?

Section 2.1 “Druggability analysis revealed a total of 12 proteins as drug targets as these twelve proteins revealed similarity with the FDA-approved drugs of the drug bank database (Table 3).” Targets are similar to FDA-approved drugs? From the table 3 I see You assessed the drugabillity of the targets by comparing them to DrugBank present targets, correct? How did You assess the druggability? How did you compare?

“Analysis of non-homology with gut flora is essential for the identification of

orthologs in gut flora. Around 1014 microorganisms reside in the gastrointestinal tract of

healthy humans [14]. Furthermore, a total of four proteins were identified as non-gut flora

proteins (Table 4). “

Explain in more detail how was this step performed. Again, from the sequences of gut-flora microorganisms and BLAST?

Section 2.2 “Among the four-drug targets that are identified, we further performed artificial intelligence-based virtual screening for beta keto acyl -ACP synthase III data were available

only for this target.” What? Improve the language by native English lector, and please explain sentence by sentence slowly what was performed….

Section 2.2

A total of 386 compounds were retrieved from the binding databank database.” Provide the download link as well as references for the reader. Provide SMILES and codes of compounds in the supporting info. On what target are the compounds active? Provide activity data.

“A total of 500 decoys were generated. B” How were the decoys generated? Provide the decoy SMILES in the supporting information.

Elaborate on the structure of the target and the dataset. How were they retrieved and prepared?

“A recursive feature selection is a technique that selects a subset of relevant features

(columns) from a dataset. A machine-learning algorithm with fewer features will run

more efficiently with less space or time complexity [19].” Elaborate on the details of the work !!! Complete KNN SVM text cannot be understood. What You actually did? Please provide code for these two models and data in Git Hub/GitLab so the reader can test them.

Docking section “The docking scores of the hits were good as compared to the

standard drug as revealed by the post-molecular docking analysis (Table 6)."

What? Provide reference compounds along with literature references. What was docked, what target etc…

The docking scores of the hits were good as compared to the standard drug as revealed by the post-molecular docking analysis (Table 6)." Where does toxicity assessment come from?

Please elaborate a bit more on how Your initial model led to the selection of compounds. Which compounds were docked and why? Do you also claim affinity calculations? What experiment for affinity calculations was performed?

Can You provide any experimental evaluation? What about other targets You identified?

The language should be lectored by a native English lector to improve the quality of language.

Reviewer 2 Report

Is Y. pestis 91001 a reference strain? I could not find it in database. No accession number provided to access.

It is incorrect to state that bubonic plague can develop into pneumonic or septicemic plague if not diagnosed and treated at a time. While it is possible for bubonic plague to progress to septicemic plague, it does not typically progress to pneumonic plague.

While several in silico studies have already been conducted on subtractive genomics portion of Y. pestis, that portion is redundant. Only novelty left is machine learning. Authors could have taken the ACP synthase directly and followed the screening. The text mentions using the BindingDB database to retrieve a dataset of compounds with inhibitory activity against beta keto acyl ACP synthase. However, it does not specify how the DUDE database was used for generating decoys. It would be helpful to provide more details on the methodology or references to support this step.

It would be more accurate to specify whether the dataset was balanced (i.e., equal number of positive and negative instances) or imbalanced (i.e., unequal number of positive and negative instances). This information is crucial for interpreting the performance measures accurately.

The text does not mention how features were selected or extracted from the dataset. Feature selection is an important step in machine learning, and it would be helpful to know the approach used or if any feature engineering techniques were applied.

Authors mention using Support Vector Machine (SVM) as a machine learning model but does not provide information on the specific parameters used. SVM models have various hyperparameters (e.g., kernel type, regularization parameter), and it would be beneficial to include these details to ensure reproducibility.

A better representation of PCA would be as a free-energy surface, and then structures could be extracted from minima and visualized, compared, and discussed. Furthermore, the cross-correlation analysis would be better done from the full simulation data rather than from he PCA analysis, and discussed, visualized and analysized. Other quantitative analyses of the dynamics could also be performed.

Authors state, ‘These results show that the beta-keto acyl ACP synthase\SANC00450 system gained less motion in phase space and clustered into a narrow phase space, indicating the system's stability (Figure 9). Fig. 9 is of methodology and this figure is missing.

Out of 11 compounds, three did well. The initial library listed at mentioned website has >1000 compounds, The initial number mentioned is too small. What criteria for it? Consider adding other small molecule libraries as well.

None

Reviewer 3 Report

Dear Authors,

Your present manuscript has been reviewed,

This Article deserves attention since it highlights a very important and new topic related to the use of different in silico skills and technics merged with Artificial Intelligence (AI) for the prediction of drug targets and their inhibitors in Yersinia pestis.

The article is well presented and written in English language. Kindly find below some of my remarks (Majors and Minors).

01- Authors are invited to add lines numbers for the manuscript.

02- In the Whole manuscript, Authors are invited to put the name of the bacteria "Yersinia pestis" followed by (Y. pestis), then in the rest of the manuscript they are invited to use Y. pestis instead of Yersinia pestis.

03- In the Whole manuscript, Authors talk about MD simulations, they are kindly invited to put at least a brief description about it.

04- In the Abstract section, last sentence, Authors are invited to replace "Yersia Pestis" by "Yersinia pestis".

05- In the Keywords section, Authors are invited to add the term "Yersinia pestis"

06- In the Introduction section, When authors talked in the first two sentences authors are invited to add, after the word "Europe", the following reference: 

Overview of Yersinia pestis Metallophores: Yersiniabactin and Yersinopine

07- In the Introduction section, A very important point, Why authors does not talk about the main virulence factors in Y. pestis. I think it would be a big plus for the article to talk, in brief, about the main virulence factors of this bacterium.

08- In the Introduction section, Last paragraph, when authors talked about the importance of in silico studies in the discovery of drug targets, Authors are invited to cite this reference:

Chelating mechanisms of transition metals by bacterial metallophores “pseudopaline and staphylopine”: A quantum chemical assessment

09- In the Results and Discussion section, Authors are invited to put all the mentioned information in the first paragraph in a Figure.

10- In the Results and Discussion section, Authors are invited to put the legends of table 1 on the same page with the table.

11- In the Results and Discussion section, Page 7, below the Figure 2, Authors start the sentence with the term "Table 97.", They are invited to explain what do they mean by this term?

12- In the Results and Discussion section, Page 10, Authors are invited to remove "Table 6. continued".

Best Regards,

Some sentences need paraphrasing.